# *Salmonella* Hessarek Gastroenteritis with Bacteremia: A Case Report and Literature Review

**DOI:** 10.3390/pathogens9080656

**Published:** 2020-08-15

**Authors:** Pierangelo Chinello, Guido Granata, Vincenzo Galati, Fabrizio Taglietti, Simone Topino, Emanuela Caraffa, Carolina Venditti, Nazario Bevilacqua, Lucilla Sbardella, Stefano Bilei, Nicola Petrosillo

**Affiliations:** 1National Institute for Infectious Diseases “L. Spallanzani”, Via Portuense, 292, 00149 Rome, Italy; guido.granata@inmi.it (G.G.); vincenzo.galati@inmi.it (V.G.); fabrizio.taglietti@inmi.it (F.T.); simone.topino@inmi.it (S.T.); emanuela.caraffa@inmi.it (E.C.); carolina.venditti@inmi.it (C.V.); nazario.bevilacqua@inmi.it (N.B.); nicola.petrosillo@inmi.it (N.P.); 2Microbiology Unit, Policlinico Casilino, 00169 Rome, Italy; lsbardella.polcas@eurosanita.it; 3Istituto Zooprofilattico Sperimentale del Lazio e della Toscana “M. Aleandri”, 00178 Rome, Italy; stefano.bilei@izslt.it

**Keywords:** salmonellosis, *Salmonella* Hessarek, bacteremia

## Abstract

*Salmonella enterica* subspecies *enterica* serotype Hessarek (*Salmonella* Hessarek) is considered a serovar with high host specificity and is an uncommon cause of disease in humans; no cases of *S.* Hessarek bacteremia have been reported in humans to date. On 16 July 2019, a young male presented abdominal pain, vomit, diarrhea, and fever up to 41 °C, a few hours after a kebab meal containing goat meat; he went to the Emergency Room, where a Film Array^®^ GI Panel (BioFire, Biomerieux Company, Marcy-L´Étoile, France) was performed on his feces and results were positive for *Salmonella*. The culture of the feces was negative, but the blood culture was positive for *Salmonella spp*., which was identified as *Salmonella* Hessarek by seroagglutination assays. The patient was treated with ceftriaxone 2 g intravenously qd for 8 days; he was discharged in good general conditions, and ciprofloxacin 500 mg per os bid for 7 more days was prescribed, after exclusion of endocarditis and of clinical signs of complicated bacteremia. This case of *Salmonella* Hessarek gastroenteritis with bacteremia is probably the first case of bloodstream human infection due to this agent ever described. Further studies are needed to ascertain the global burden of *S.* Hessarek disease in humans.

## 1. Introduction

*Salmonella enterica* subspecies *enterica* serotype Hessarek (*Salmonella* Hessarek) belongs to the serogroup B; it is considered a serovar with high host specificity, and it is an uncommon cause of disease in humans; few cases or outbreaks of *S.* Hessarek human infection are reported [1,2,3,4]. *S.* Hessarek was first detected from a Common Raven (*Corvus corax*) in Iran in 1953 [5]; outbreaks of septicemic salmonellosis in wild birds and in European mammals, possibly related to ingestion of infected birds, have been described [3,6,7,8,9,10]. To the best of our knowledge, no cases of *S.* Hessarek bacteremia have been reported in humans to date. We describe a case of *S.* Hessarek gastroenteritis with bacteremia in a young man.

## 2. Case Presentation

On 16 July 2019, a 30-year-old male patient presented abdominal pain, vomit, diarrhea, and fever up to 41 °C, a few hours after a kebab meal containing goat meat. On 17 July, he went to the Emergency Room of the *Policlinico Casilino* Hospital in Rome (Italy), where a Film Array^®^ GI Panel (BioFire, Biomerieux Company) was conducted on his feces and the results were positive for *Salmonella*. The fecal specimen was seeded, after enrichment in selenite broth (BBL Selenite Becton-Dickinson and Spark company, MD, USA), on “Chromagar Salmonella” and “Hektoen Enteric Agar” (Becton-Dickinson GmbH, Heidelberg, Germany). The cultures were incubated for 24–48 h at 37 °C. Moreover, a set of Bactec Plus aerobic broth blood cultures (Becton-Dickinson Inc., Sparks, MD) was sent to the Microbiology Laboratory. After two days of incubation at 37 °C in a Bactec 9420 automated culture system, the blood culture tested positive. Microscopic examination conducted with Gram stain revealed the presence of Gram-negative bacilli, confirmed by growth on Blood Agar, Mac Conkey Agar (Becton-Dickinson GmbH, Heidelberg, Germany) after incubation at 37 °C for 24 h. The identification of *Salmonella spp* was carried out, following manufacturer’s instructions, with matrix-assisted laser desorption/ionization flight time mass spectrometry (MALDI-TOF TECHNIQUE, Bruker Daltonik GmbH, Bremen, Germany) performed in duplicate on a single colony—score values ≥2.0 were accepted.

The culture of the feces was negative, but the blood culture was positive for *Salmonella spp*., whose antibiogram showed sensitivity to ampicillin (minimum inhibitory concentration (MIC) ≤2 mg/L), cefotaxime (MIC ≤1 mg/L), levofloxacin (MIC ≤0.5 mg/L) and cotrimoxazole (MIC ≤1 mg/L for trimethoprim and ≤19 mg/L for sulfamethoxazole); the MICs were determined by Phoenix System (Becton-Dickinson Diagnostics, CA, USA); the strain was also sensitive to nalidixic acid according to the Clinical and Laboratory Standards Institute (zone diameter 21 mm, Kirby Bauer), but was resistant to aminoglycosides (gentamicin and tobramycin MIC = 2 mg/L). The *Salmonella* grown on blood was sent to the *Istituto Zooprofilattico Sperimentale del Lazio e della Toscana* in Rome (Italy) where it was identified as *Salmonella* Hessarek by seroagglutination assays (SSI Diagnostica, Denmark) and the antigenic formula was 4,12:a:1,5.

On 19 July the patient was transferred to the *“L. Spallanzani”* National Institute for Infectious Diseases in Rome (Italy). On admission he was alert, eupnoic, and dehydrated; the blood tests showed the following: white blood cells 5270/mmc (polymorphonuclear cells 56%), hemoglobin 16.1 g/dL, platelets 226,000/mmc, C reactive protein 5.67 mg/dL, aspartate aminotransferase 24 U/L, alanine aminotransferase 36 U/L, creatinine 1 mg/dL; the abdominal ultrasound was unremarkable, apart from hepatic steatosis. The patient was initially treated with intravenous rehydration; diarrhea and fever rapidly subsided and his general conditions improved. When the result of the blood culture was known, the patient was given ceftriaxone 2 g intravenously qd for 8 days; the transthoracic echocardiography showed no signs of endocarditis. The patient was discharged on 1 August 2019 in good general conditions, the survey fecal and blood cultures were negative, and ciprofloxacin 500 mg per os bid for 7 more days was prescribed. After a 9-month follow-up, the patient was in good health conditions and no relapses of salmonellosis occurred. Two other friends (21 and 28 years old, respectively) of our patient were admitted at “L. Spallanzani” Institute on 19 and 20 July 2019 with fever, diarrhea and vomit, after having shared the same kebab meal. They both had a Film Array^®^ GI Panel test positive for *Salmonella* on their feces, but the culture of their feces and blood tested negative. They rapidly recovered from their intestinal infection and are currently in good general conditions.

## 3. Discussion

The genus *Salmonella* includes approximately 2500 serotypes and host specificity is known in some of these serotypes. *S.* Hessarek has an apparently high specificity and pathogenicity for Starlings (*Sturnidae)* and Song Thrushes (*Turdus philomelos)*, but other susceptible species of birds are known, including the House Sparrow *Passer domesticus*, the Eurasian Blackbird (*Turdus merula)* and White Wagtail (*Motacilla alba)* [3,6]. *S.* Hessarek was also isolated from mammals, including foxes, pigs, and a lynx [8,9,10]; these mammals could have been infected by the ingestion of infected birds, as is suggested in the case of *S.* Typhimurium in foxes and cats [9,11]. *S.* Hessarek was also recently detected in wild boars, but not in farmed pigs, in an area of northern Italy (Emilia-Romagna) [12].

*S.* Hessarek was first associated with disease in humans in Israel in 1950s [3,13], but the body site of isolation was not reported. In 2005, the Australian Capital Territory reported five human cases of *S.* Hessarek infection caused by contaminated eggs; the eggs were served at a restaurant as poached eggs and as hollandaise sauce; again, the specimen of isolation was not specified [2]. A protracted outbreak of 25 cases of human *S.* Hessarek gastroenteritis was described in South Australia from March 2017 to July 2018. Of note, the infection was associated with one brand of eggs and *S.* Hessarek was isolated in the content of eggs, but not in the egg-shell rinse [1], suggesting the need of further research to understand the behavior of *S.* Hessarek, including its ability to be present inside the eggs. When looking for *S.* Hessarek bacteremia in humans, we were not able to identify any previously reported case (Table 1).

With 91,857 confirmed cases, salmonellosis was the second most commonly reported gastrointestinal infection in humans after campylobacteriosis in the European Union (EU) in 2018, resulting in an EU notification rate of 20.1 cases per 100,000 people. The highest notification rate was reported by Slovakia (124.8 cases per 100,000 inhabitants), while the lowest rates were reported by Cyprus, Greece, Italy, and Portugal (≤ 6 cases per 100,000 inhabitants). In Italy 3656 cases of human salmonellosis were described in 2018, with a decreasing trend from 2009 to 2018. *S.* Hessarek was not in the top 20 list described in the 2019 European Food Safety Authority report [14]. The main sources of human *Salmonella* infection include eggs and eggs product, poultry meat, and pork. Our patient and his friends had a kebab meal containing goat meat before the beginning of their symptoms. Goat meat has been described as a potential source of *Salmonella* serovars associated with human disease [15].

The strain of *S.* Hessarek isolated in this patient was resistant to aminoglycosides (gentamicin and tobramicin MIC = 2 mg/L) and sensitive to ampicillin, cephalosporins, and fluoroquinolones. *Salmonella spp.* resistance to aminoglycosides has been frequently ascertained in Italy: in a national survey, 45% of *Salmonella* strains were resistant to gentamicin [16]. However, gentamicin has limited activity against intracellular pathogens, so even if some isolates may appear susceptible in vitro, gentamicin cannot be relied on in vivo [17]. The patient was treated with an 8-day course of intravenous ceftriaxone followed by ciprofloxacin 500 mg twice daily for 7 more days, after exclusion of endocarditis and of clinical signs of complicated bacteremia. Patients diagnosed with non-typhoid *Salmonella* bacteremia typically need a 7–14-day antibiotic course [18]; however, more prolonged antimicrobial therapy has been recommended in the case of complicated bacteremia [19].

In conclusion, we have reported a case of *Salmonella* Hessarek gastroenteritis with bacteremia, which is probably the first case of bloodstream human infection due to this agent ever described. Further studies are needed to ascertain the global burden of *S.* Hessarek disease in humans. This strain is deposited in the collection, and subsequently can be investigated in whole-genome sequencing and used for comparative genomics, since in the future the importance of this serovar in the epidemiology of infection in EU may increase.

## Figures and Tables

**Table 1 pathogens-09-00656-t001:** Cases of *Salmonella* Hessarek infection in humans.

Reference	Years	Country	Likely Source	Number of Cases and Infection Site
[3]	1952–1954	Israel	Unspecified	3 cases, unspecified infection site
[2]	2005	Australia	Eggs	5 cases, gastrointestinal
[1]	2017–2018	Australia	Eggs	25 cases, gastrointestinal
[4]	2009–2016	Australia	Chicken, eggs * and unspecified	140 cases, unspecified infection site
Present case	2019	Italy	Kebab with goat meat	1 case, gastrointestinal + bacteremia

*: see reference 1 for likely sources in 2014 and 2016 outbreaks.

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
