# Peer review of "Salmonella Hessarek Gastroenteritis with Bacteremia: A Case Report and Literature Review"

_pathogens, 2020, doi:10.3390/pathogens9080656_

Round 1
Reviewer 1 Report
Authors presented a medical case report of specific serovar Salmonella enterica subspecies enterica serotype Hessarek, or simply Salmonella Hessarek, where human infection has never been identified before. Salmonella Hessarek is not totally new, this serovar has been known to the scientific field and known to cause infection in birds and maybe animals.
The paper is generally well written and language is good. This is the first report that S. Hessarek serovar can cause human infection and the antibiograms have been reported. Two potentially linked infection cases were reported though not confirmed infection with the same serovar. This discovery can have value to clinical practitioners and people who specialised in Salmonella. As yet, the serovar has not been a major issue for medical system, but this report present an additional question if this serovar can be serious burden for medical system in the future.
The authors presented a single case of Salmonella infection of a serovar Hessarek, which is very rare to see in human, and this may be the 1st blood sample confirmed case. This is typical clinical style of publication and is of some value to clinical community. However, in Section 2 Case presentation, this patient's faecal sample was tested positive for Salmonella by Film Arrray GI Panel but negative for faecal culture. This raises questions of the procedure of the culturing, i.e. if broth culturing was used and how it was performed. It is not impossible to have positive results by Array without culturing but negative results by culturing. But this needs to be explained better or at least authors need to describe clearly how the culturing experiment was performed to explain this contradicting results. Besides, all the blood culturing and other procedures,little details were given by authors. And for the integrity of scientific publication, these details are important.
Reviewer 2 Report
The manuscript “Salmonella Hessarek gastroenteritis with bacteraemia: a case report and literature review” desсribes the first case of human bacteremia due to rarely isolated Salmonella serovar Hessarek and a comprehensive literature review about this serovar. Since the manuscript was written by non-English-speaking authors, I would highly recommend English editing. Nevertheless, I have not found any major problems in this manuscript. It would be nice to know any molecular genetic characteristics of the strain, perhaps that is a topic for a future publication?
My minor points are the next ones:
Line 37: Can the authors give a more exact patient's age, because "young" is a very vague term. The same goes for the other two cases mentioned below.
Line 41: Which method was used to determine MIC for the antibiogram? And what is ≤1/19 mg/L for cotrimoxazole?
Line 45: Salmonella should be in italic.
Line 46: Who is the serum manufacturer? Also, it would be good to put here the antigenic formula for Salmonella Hessarek.
Line 59: Could you provide a clinic and symptoms for the other two friends which were admitted to the “L. Spallanzani” Institute?
Line 61: Salmonella should be in italic.
Lines 67-68: Please, provide the Latin names for Starlings, Song Thrushes, Eurasian Blackbird, and White Wagtail.
Line 92: Salmonella should be in italic.
Line 96: It was not mentioned above (lines 41-43) that the strain was resistant to aminoglycosides (gentamicin and tobramycin MIC = 2 mg/L).
Line 110: Could you provide more clear authors' contribution, especially there are 12 authors in the manuscript? Obviously, only one or two of them wrote the text, and so on. Moreover, it is not shown who is the corresponding author for the manuscript.
Also, it will be good to write in conclusion that this strain is deposited in the collection, and subsequently can be investigated in whole-genome sequencing and used for comparative genomics, since in the future the importance of this serovar in the epidemiology of infection in EU may increase.
